# Forecasting Urban Water Demand Using Cellular Automata

**Laís Marques de Oliveira \*, Samíria Maria Oliveira da Silva** 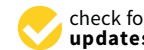**, Francisco de Assis de Souza Filho, Taís Maria Nunes Carvalho** **and Renata Locarno Frota**

Hydraulic and Environmental Engineering Department (DEHA), Federal University of Ceará, Fortaleza 60020-181, Brazil; samiriamaria@gmail.com (S.M.O.d.S.); assissouzafilho@gmail.com (F.d.A.d.S.F.); taismarianc@gmail.com (T.M.N.C.); renata.locarno@hotmail.com (R.L.F.)

\* Correspondence: lais.engciivil@gmail.com

**Abstract:** Associating the dynamic spatial modeling based on the theory of cellular automata with remote sensing and geoprocessing technologies, this article analyzes what would be the per capita consumption behavior of Fortaleza-CE, located in the Northeast of Brazil, in 2017, had there not been a period of water scarcity between 2013 and 2017, and estimates the future urban water demand for the years 2021 and 2025. The weight of evidence method was applied to produce a transition probability map, that shows which areas will be more subject to consumption class change. For that, micro-measured water consumption data from 2009 and 2013 were used. The model was validated by the evaluation of diffuse similarity indices. A high level of similarity was found between the simulated and observed data (0.99). Future scenarios indicated an increase in water demand of 6.45% and 10.16% for 2021 and 2025, respectively, compared to 2017. The simulated annual growth rate was 1.27%. The expected results of urban water consumption for the years 2021 and 2025 are essential for local water resources management professionals and scientists, because, based on our results, these professionals will be able to outline future water resource management strategies.

**Keywords:** water demand; cellular automata; dynamic modeling

## 1. Introduction

Growing water demand in urban centers around the globe can be attributed to growing population and urbanization rates, given limited water resource availability. Limited water resources increase water use conflicts and reduce access to good quality water, making urban supply a major challenge for water resource management [1].

In addition to population growth, the increase in water demand may also be due to per capita consumption growth. Besides climate, some of the main factors driving per capita water consumption are: (i) technical-hydrometric network, water tariff, dynamic and static pressures in the distribution network, water quality; (ii) socioeconomic factors such as per capita income [2].

Prior researches corroborate this assertion. Rodrigues and Garcia [3] evaluated the correlation between socioeconomic and structural variables (number of supplied inhabitants, water supply network extension, average water tariff, average annual rainfall and temperature, GDP and per capita income) and water consumption in Brazil, using linear regression. The number of people attended by the water supply, the municipal GDP, water network extension and the per capita income presented high explanatory power, with $R^2$ ranging between 0.470 and 0.859.

In Brazil, several studies analyzed water consumption in different spatial scales. Authors such as Chaib et al. [4] and Narchi [5] discussed the factors influencing water consumption within households, while Fernandes Neto [6], Guedes et al. [7] and Dias [8] evaluate water consumption in Minas Gerais,

in northeast region and different areas of Belo Horizonte (MG), respectively. Despite the importance of looking at present water consumption in Brazilian cities, these studies only analyze current water demands and, therefore, do not provide any insight into water availability subject to urban expansion. Our contribution is to include projections of future water demands.

The management of urban water demand becomes urgent in urban centers in arid/semi-arid regions of developing countries (the case of Northeast Brazil), where climatic conditions, combined with obsolete supply systems and the inadequacy or absence of urban planning sensitive to water resources, determine the emergence of serious water supply problems, making it challenging to meet the quantitative and qualitative demands of the population [9,10].

One of these problems is related to obsolete supply systems that present deteriorated water infrastructure, causing water losses, such as leaks. Detecting the cause of these leaks is essential for sustainable management.

Meniconi et al. [11] wrote about the use of innovative techniques for detecting faults in the water distribution network that cause the leaks. They claim that the most common fault detection technologies are those that use a line with sensors inserted in the ducts. A numerical model evaluates the leaks, and this model represents the layout of the water distribution system, considering all the conditions of the operation of this system.

Given the importance of studies on water consumption in Brazilian cities for water resource planning, it is important to understand the behavior of future water demands in addition to current water consumption, especially in areas that are subject to urban expansion. To predict the water demand for the city of Fortaleza, Brazil, we applied dynamic spatial modeling based on the theory of cellular automata, with remote sensing and geoprocessing technologies. With this model, we were able to simulate the per capita consumption prospects for 2021 and 2025, and present the result in this article.

The association of remote sensing and geoprocessing technologies with dynamic spatial models used in the urban analysis allows a quantitative assessment of the structuring and dynamics of urban space, provides a better visualization of the urban reality and the elements responsible for its spatio-temporal transformations [12]. One element responsible for transformations taking place in the urban space is the demand for urban water.

Estimating water demand of urban centers is crucial for the planning and management of cities and water supply companies. The spatialization of water demand is useful for the decision-making process, especially during a crisis, when it is necessary to set water supply priorities. Hence, assessing the spatial distribution of the increase in water consumption rates is an excellent tool for water resources management in urban areas [13].

## 2. Water Demand Prediction

Water demand prediction plays a vital role in the optimization of water supply systems operation. Accurate forecasting is necessary for the efficient allocation of limited water resources, helping avoid wasteful use of water resources due to the misallocation of water [14]. Water demand forecasting is also a vital element in urban planning and sustainable development of cities [15], helping policymakers, with many important decisions on water demand management, environmental planning and optimal use of water resources.

Forecasting water demand is essential in order to address overload problems in water supply systems worldwide, due to high population growth and increased per capita water consumption [16].

The rapid population growth makes the efficient management of water supply a challenging task [17]. According to Suhartono et al. [18], one of the most important duties for the water company is to attend consumer demand, even when it implies energy waste and financial problems for the company.

Therefore, a method for accurate and efficient forecasting is necessary for sustainable and economic management planning. It is essential to know that the difference between dry and rainy periods will affect water consumption in the community, indicating a trend and a seasonal pattern in the water

demand data, and researchers are developing studies using intelligent technologies, such as neural networks and cellular automata studies.

Giacomoni and Berglund [19], for example, simulated adaptive management of water demand through the development of a complex framework, combining cellular automata, agent-based and hydrological modeling to simulate changes in land use, consumer behavior, management decisions, and the reservoir flow and storage processes. The results demonstrated that adaptive demand management strategies motivated by water scarcity resulted in the long-term reduction of per capita water demand.

Shah et al. [20] proposed a differential learning model based on a neural network to model over- and under-consumption. Although the model did not considerably reduce the forecast error for days with average water demand, it did provide lower and upper limit estimates for water demand for atypical days.

Using the cellular automata theory, Oliveira [21] simulated future scenarios of land use and verticalization of two neighborhoods in Campina Grande, Brazil, based on changes detected between 2011 and 2018. Future water demand needs of the population were estimated through future land use and verticalization scenarios (2040, 2070 and 2100).

We also believe that Cellular Automata (CA) is a powerful approach to model complex dynamic systems such as urban systems. It is a technique that, according to Votsis [22], has the advantage of simulating the city evolution concomitantly with the impacts of spatial interventions, as well as the ability of spatially reproduced growth distribution.

## 3. Study Area

Fortaleza, the capital of Ceará, is in the Northeast region of Brazil, Figure 1. According to the Brazilian Institute of Geography and Statistics (IBGE) [23], Fortaleza has a population of 2,643,247 inhabitants and an area of 312.407 km$^2$.

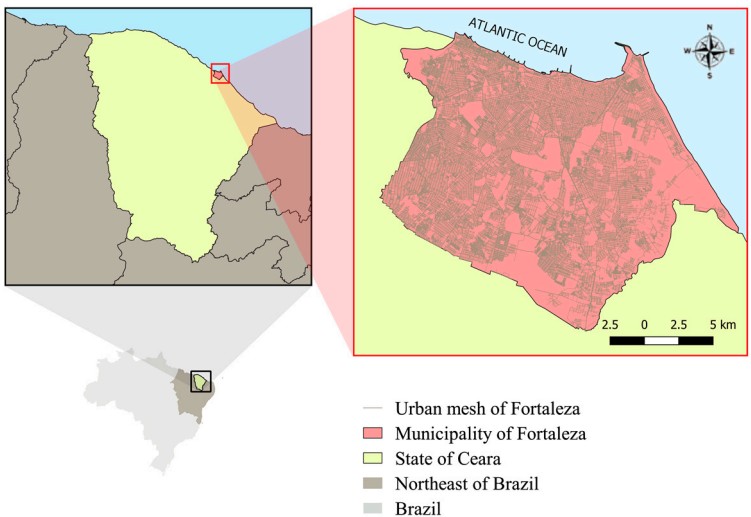

**Figure 1.** Study area location.

According to the Municipal Secretariat of Urbanism and Environment of Fortaleza (SEUMA), historically, the water supply of the RMF has suffered crises where water availability was insufficient to serve the population, needing to import water from other hydrographic basins in the State of Ceará. The last major supply crisis occurred in 1993, when it was necessary to cross waters of the Jaguaribe River hydrographic basin, through the Canal do Trabalhador, a work carried out under a water emergency in the state [24].

Fortaleza's raw water supply is composed of local and interregional waters from the Jaguaribe-Metropolitano system [25]. The local supply is stored in five reservoirs that have a total capacity

of 871 hm$^3$, namely: Gavião, Pacoti, Riachão, Pacajus, and Aracoiaba. Due to the growing demand, this system started to receive water from the Jaguaribe basin, forming the Jaguaribe–Metropolitano system.

The interregional storage system has a total capacity of 10,241 hm$^3$ distributed in three reservoirs: Orós with 1940 hm$^3$, Banabuiú with 1601 hm$^3$ and Castanhão with 6700 hm$^3$. The useful storage of Castanhão (deducting flood waiting volume) for urban, irrigation, and industrial uses is 8002 hm$^3$, and 56% of the capacity of Jaguaribe–Metropolitano comes from this reservoir.

The raw water is directed to the water supply system (WSS), which carries out the treatment and distribution of treated water in the city. The WSS is composed of works for water capitation, adduction, treatment, storage, pumping, and distribution.

## 4. Data and Methods

In order to predict the water demand of Fortaleza, future scenarios of per capita water consumption were simulated. The simulation was performed with geotechnology techniques and cellular automata theory, and divided into three steps: data acquisition, spatialization of data in matrices, and finally, dynamic modeling, Figure 2. The water micro-metering data, which refer to data on user consumption, was collected from the Ceará Water and Wastewater Company (CAGECE) for the period between 2009 and 2017.

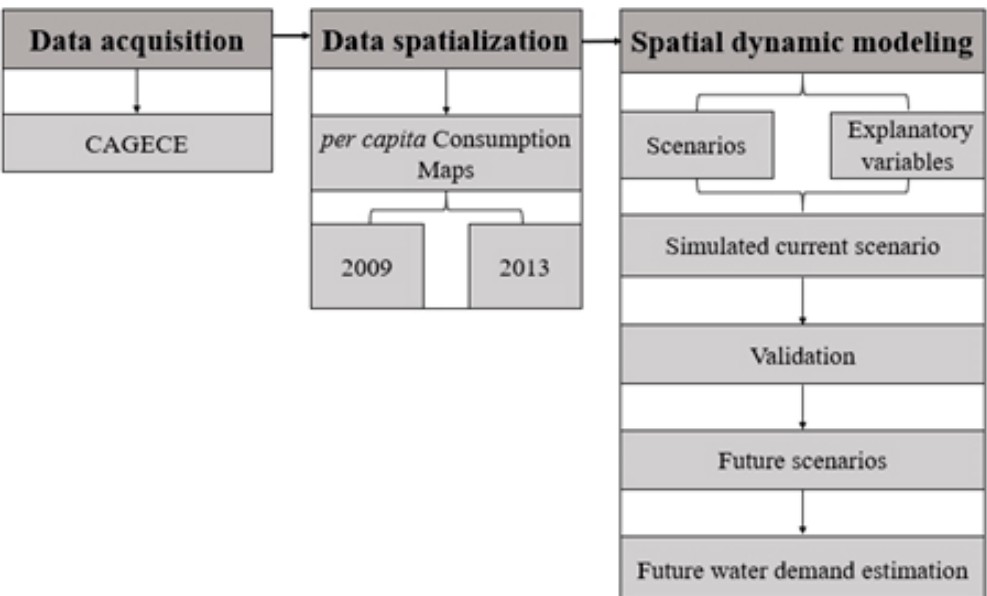

**Figure 2.** Schematic diagram of the methodological steps.

The CA-based model was carried out in the Dinamica EGO software (an acronym for Environment for Geoprocessing Objects), developed by the Remote Sensing Center of the Federal University of Minas Gerais (CSR/UFMG). Data spatialization was performed in QGIS software.

### 4.1. Input Data for the Simulation Model

In this study, two urban water consumption scenarios were applied to compose the input data of the simulation model: (i) Scenario I: 2009; and (ii) Scenario II: 2013. These periods were chosen because they represent average water supply conditions for Fortaleza. There was an increase in total consumption from 2009 to 2013, with a declining trend starting in 2014. In the first period (2009–2013), growth was 19%, between 2013 and 2017, the decline was 26% in the consumption, with a 12% drop only between 2016 and 2017 (Table 1). It is worth remembering that the state faced a period of water scarcity crisis between the years 2013 and 2017, contributing to this reduction in consumption that occurs since 2014, which mainly affects the availability of the Jaguaribe–Metropolitano system.

Consumption data were grouped by census tracts (n = 2952) into 6 classes: Class 1:0 to 50 L/inhabitant/day; Class 2:50 to 200 L/inhabitant/day; Class 3:200 to 400 L/inhabitant/day; Class 4:400 to 600 L/inhabitant/day; Class 5:600 to 800 L/inhabitant/day and Class 6: above 800 L/inhabitant/day. Data were spatialized, generating a categorical map of per capita consumption.

Explanatory variables (static and dynamic) of the changes observed in water consumption were also used as input data. "Map of difference in water consumption between the years 2009 and 2013" and "Distance to water consumption classes" were the static and dynamic variables, respectively.

**Table 1.** Distribution of volume (m$^3$) and percentage of consumption by category, 2009–2017.

| Year | Consumer Category | | | | Total |
|------|-------------|------------|------------|--------|-------|
|      | **Residential** | **Commercial** | **Industrial** | **Public** | |
| 2009 | 106,478,863 | 6,835,526 | 3,476,270 | 4,346,633 | 121,137,292 |
|      | 87.9% | 5.6% | 2.9% | 3.6% | 100.0% |
| 2010 | 115,284,964 | 7,569,662 | 3,809,749 | 4,731,477 | 131,395,852 |
|      | 87.7% | 5.8% | 2.9% | 3.6% | 100.0% |
| 2011 | 113,630,513 | 7,820,580 | 3,905,774 | 4,565,486 | 129,922,353 |
|      | 87.5% | 6.0% | 3.0% | 3.5% | 100.0% |
| 2012 | 116,080,193 | 8,304,057 | 4,893,166 | 4,974,884 | 134,252,300 |
|      | 86.5% | 6.2% | 3.6% | 3.7% | 100.0% |
| 2013 | 125,176,439 | 9,297,287 | 4,860,388 | 5,061,548 | 144,395,662 |
|      | 86.7% | 6.4% | 3.4% | 3.5% | 100.0% |
| 2014 | 119,355,633 | 8,429,656 | 4,401,631 | 5,039,227 | 137,226,147 |
|      | 87.0% | 6.1% | 3.2% | 3.7% | 100.0% |
| 2015 | 112,524,506 | 8,039,630 | 4,024,938 | 4,890,770 | 129,479,844 |
|      | 86.9% | 6.2% | 3.1% | 3.8% | 100.0% |
| 2016 | 104,562,627 | 7,056,524 | 4,730,244 | 4,573,362 | 120,922,757 |
|      | 86.5% | 5.8% | 3.9% | 3.8% | 100.0% |
| 2017 | 90,209,543 | 6,409,223 | 3,954,204 | 5,742,885 | 106,315,855 |
|      | 84.9% | 6.0% | 3.7% | 5.4% | 100.0% |

*4.2. Model Calibration and Validation*

Model calibration consists of four steps: calculating transition matrices; calculating weights of evidence coefficients; map correlation analysis and adjustment and execution of the simulation model. The transition matrix provides the percentage of land that will change from one class to another. This information is obtained by cross-tabulation between the initial (2009) and the final (2013) class maps [12].

As the input maps (2009 and 2013 maps) are categorical, that is, they have states defined by numbers, the Dinamica EGO cellular automata system understands these numbers as representing a state of a given cell this case is the pixel. Then, when calculating the transition matrix, the automata can identify how many pixels in state 1 in 2009 changed to state 2 in 2013.

Based on the initial and final maps, we calculated the transition rates in a single step (total period of 4 years) and multiple steps (annualized). From this, it was possible to generate, respectively, a transition matrix with the changes that occurred in the total of 4 years between the years 2009 and 2013 and four transition matrices with annual changes: (i) from 2009 to 2010, (ii) 2010 to 2011, (iii) 2011 to 2012 and (iv) 2012 to 2013. All these transition matrices are derived from an ergodic matrix, that is, a matrix that has real eigenvalues and eigenvectors [26].

In the stage of determining the weights of evidence, the static variable and dynamic variables are inserted in the model. The difference map (static variable) was obtained by a matrix subtraction in QGIS, specifically, between the maps of the year 2009 and 2013. The distances for the consumption

classes that have undergone transitions (dynamic variables) were calculated within the simulation model of the EGO Dynamics software, using the "Calc to Distance Map" ("Distance") function. Thus, both the static variable and the dynamic enter the simulation model from the stage of determining the weights of evidence (Figure 3).

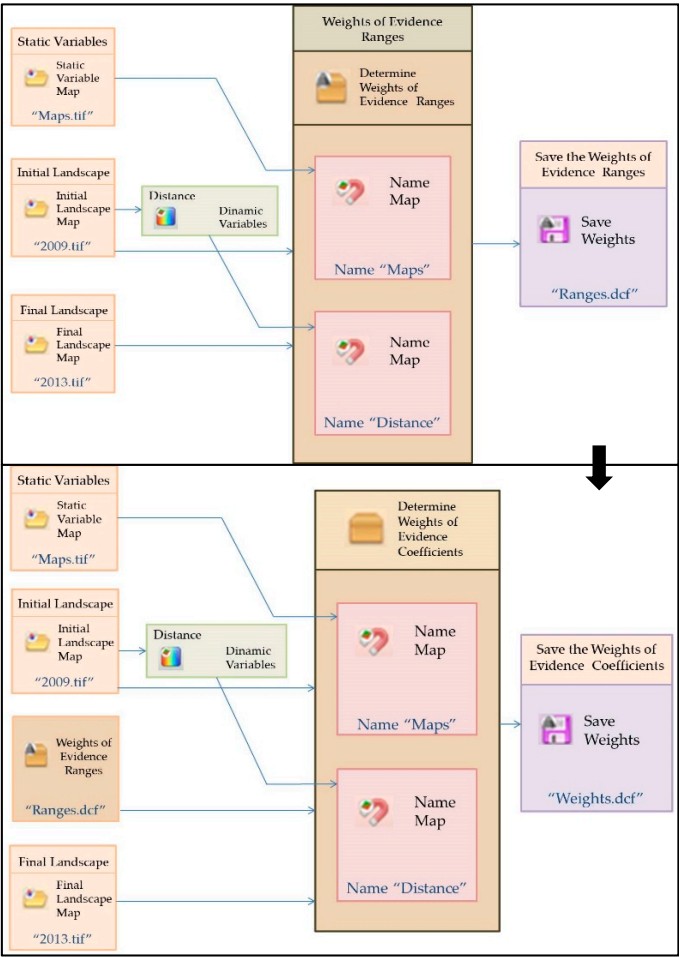

**Figure 3.** Steps for determining the Weights of Evidence.

The "Calc to Distance Map" ("Distance") function receives the categorical map corresponding to the initial year (2009), and also receives the values corresponding to the consumption classes for which it must calculate distances (Euclidean distances), based on class position information (pixels) from the 2009 map. In our simulation, there were consumption transitions for classes 1, 2, 3, 4, and 6, from 2009 to 2013, these were the numbers entered in the "Calc to Distance Map" function for calculating distances. With that, it generated five maps (in the case of our simulation) of border distance (closest distance) of the cells corresponding to consumption classes 1, 2, 3, 4, and 6, based on the 2009 map. The function identifies consumer class 1 pixels on the initial map and calculates the distances between them and the surrounding areas. Thus, the function takes into account the proximity of consumption areas 1 in relation to the probability of a new consumption class 1.

The resulting maps have their distances represented in meters. The map format has the same dimensions as the categorical map (2009 map). Thus, the Euclidean distances calculated in a two-dimensional plane are $D(pq) = \sqrt{(px - qx)^2 + (py - qy)^2}$, for two-dimensional points $P = (px, py)$ e $Q = (qx, qy)$.

The final map (2013) is not used in the "Calc Distance Map" function; it is only used in the "Weights of Evidence Ranges" and "Determine Weights of Evidence Coefficients" functions, as data to

assist in the determination of evidence weights, that is, from the 2013 map, it is verified whether these ranges of distances calculated from the 2009 map influenced a class change registered between 2009 and 2013, according to the proximity between the consumption classes (pixels). If it is proven that these distances (dynamic variables) influence the change of class by proximity, they receive an influence value (weights of evidence); the closer an area is to a particular class that influences, the greater the value that weight.

In this sense, the weights of evidence method are applied in Dinamica EGO to produce a transition probability map, which represents the most favorable areas for a change [27]. This method consists of a Bayesian method, in which the effect of a spatial variable on a transition is calculated regardless of a combined solution. Weights of evidence represent each influence on a variable in the spatial probability of a transition $i \rightarrow j$, and are calculated as follows:

$$O\{D|B\} = \frac{P\{D|B\}}{P\{\overline{D}|B\}}$$

$$\log\{D|B\} = \log\{D\} + W^+$$

where $W^+$ is the weight of evidence for the occurrence of event D, given a spatial pattern B. The a posteriori probability of a transition $i \rightarrow j$, given a set of spatial data (B, C, D, ... , N), is expressed as follows:

$$P\{i \rightarrow j | B \cap C \cap D \dots \cap N\} = \frac{e^{\sum W_N^+}}{1 + e^{\sum W_N^+}}$$

where B, C, D and N are the values of the spatial variables that are measured at the location $x$, $y$ and represented by their weights $W_N^+$.

For the application of the weights of evidence method in the simulation model, it was necessary to categorize the variable maps applied in the simulation model, since the weights of evidence are applied only to categorical data. Using both the QGIS and the difference map (between the years 2009 and 2013), it was possible to categorize the static variable. Meanwhile, the simulation model in Dynamics EGO categorized the dynamic variable, where a line generalization algorithm defined the categorical ranges [26], that is, the best fit curve delimits the categorical ranges of change from a series of straight lines segments.

The only necessary assumption for the weights of evidence method is that the input maps must be spatially independent. Then, correlation analysis was applied to analyze the spatial interdependence of the maps. We performed this analysis using two tests, the Cramer's V (V) test and the joint information uncertainty test (JIU).

According to Bonham-Carter [28], the closer V and JIU are to 1, the higher the spatial dependence between the considered pair of variables. Variables with a correlation above 0.5 (50%) should be neglected or combined, to replace the correlated pair in the model.

Model parameterization also includes the adjustment of the simulation model. Dinamica EGO uses as a local cellular automaton rule, a transition mechanism composed of two complementary transition functions: Patcher and Expander ("Updated Landscape" box) (Figure 4). The Patcher function generates new patches with a seeding mechanism, while the Expander function accounts for the expansion or contraction of previously existing patches of a given class [27].

In addition to defining the Patcher/Expander ratio, both mean and variance of the patch size and the patch isometry must be set. We applied R Studio software to calculate these variables. These parameters are used to standardize patch sizes that either appear or expand from existing ones as transformations occur. The patch isometry ranges from 0 to 2, increasing as the patches take on a more isometric form. The degree of the patch size fragmentation is inversely proportional to the isometry [29].

Thus, the Patcher function creates new patches for a given consumption class according to the trend of change, and the Expander function only expands the existing patches of a specific consumption class around it, according to an increasing trend for that particular class.

After defining the parameters, annual simulations were generated from the initial and final maps. Model performance was validated through the fuzzy similarity index, developed by the Remote Sensing Center of the Federal University of Minas Gerais (CSR/UFMG). This index is based on the fuzzy similarity index created by Hagen [30]. Two maps of difference were compared: the first obtained from the initial and observed final maps, and the second, obtained from the initial map and the simulated final map (Figure 5). An exponential decay function with a window size of $11 \times 11$ and a constant decay function, calculated with the following window sizes: $1 \times 1$, $3 \times 3$, $5 \times 5$, $7 \times 7$, $9 \times 9$ and $11 \times 11$, were adopted. The window sizes are grids of pixels, where each pixel corresponds to an area of 900 m$^2$ (30 m $\times$ 30 m).

After validating the model, two future water demand scenarios were simulated, using the same parameters defined in calibration and validation steps, changing only the number of iterations. In this step, 2021 and 2025 trend scenarios were generated.

Thus, short and medium-term intervals were adopted to define future scenarios. The 2017 scenario was simulated, in order to compare the consumption under average conditions (simulated map) with the water use after drought (observed map), since Fortaleza was affected by drought between 2013 and 2017.

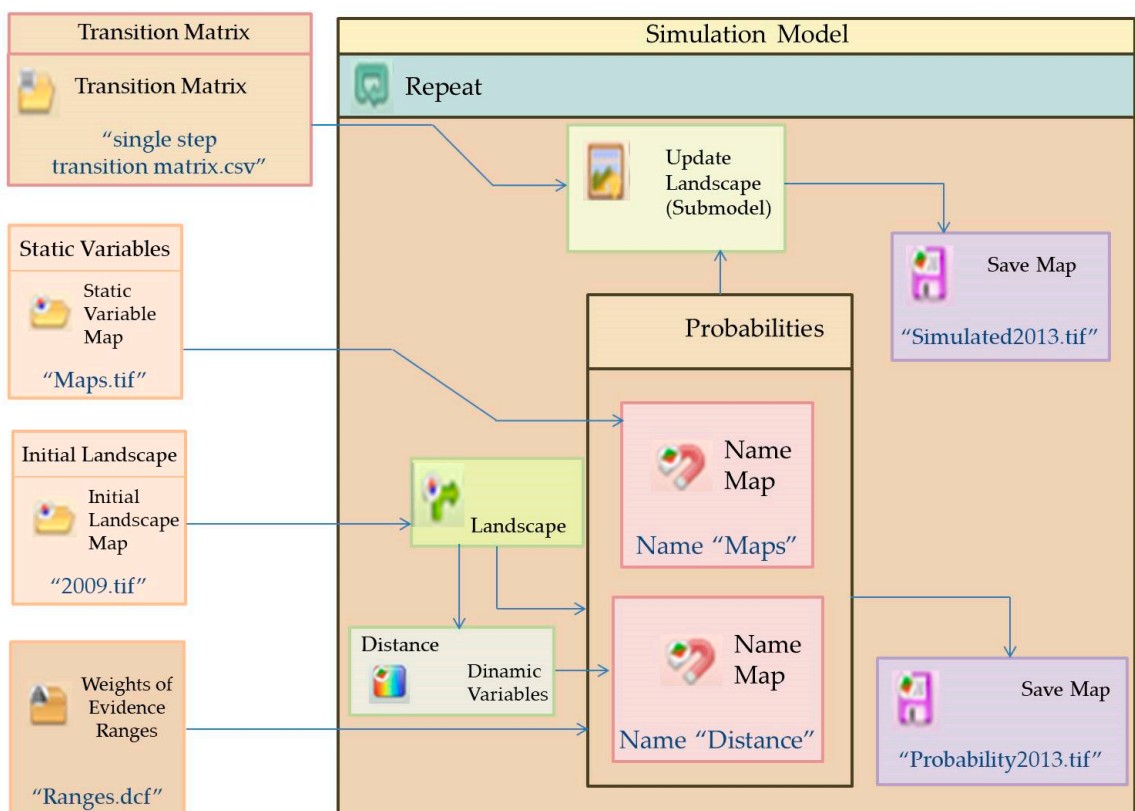

**Figure 4.** Calibration step using the Patcher and Expander functions.

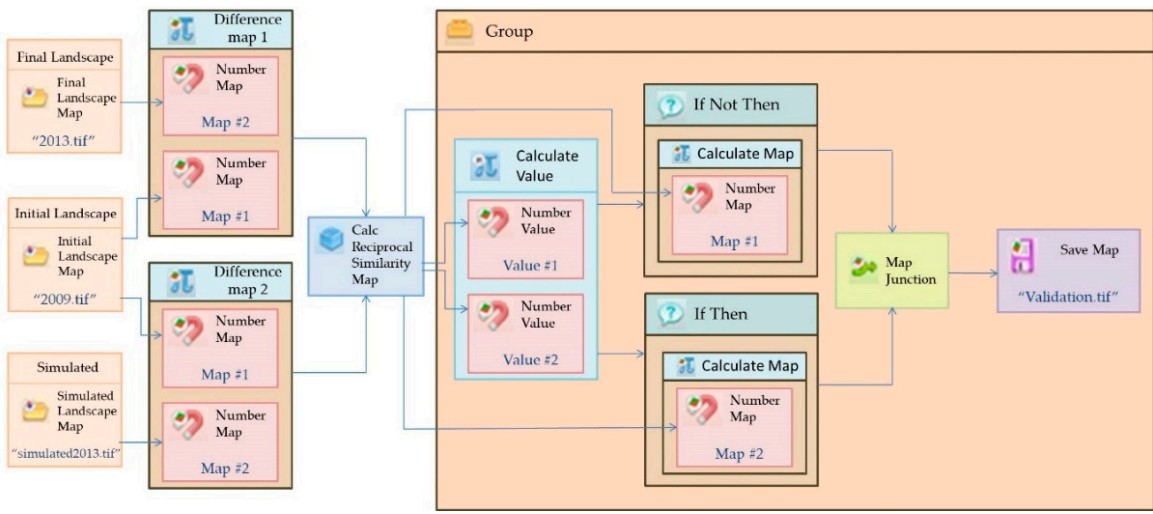

**Figure 5.** Validation step by fuzzy similarity.

## 4.3. Estimation and Validation of Future Water Demands

Future water demand was calculated by multiplying the demand/pixel rate by the number of pixels in each class of water demand. Consumption data of 2010 have been used as reference, Table 2. Thus, for scenarios 2021 and 2025, water demand was estimated using a "per pixel" rate, based on the amount of pixel for each demand class, that is, water demand rate per pixel x number of each demand class. Figure 6 shows a summary of the steps taken to estimate future water demand.

**Table 2.** Demand/pixel rate (L/day).

| Demand/Pixel Rate (2010) | | | |
|---|---|---|---|
| **Water Demand Classes** | **Number of Pixels** | **Water Demand (L/day)** | **Rate (L/day)** |
| 1 | 19,061 | 2,284,771 | 119.87 |
| 2 | 283,244 | 245,652,412.1 | 867.28 |
| 3 | 23,023 | 284,515.54 | 1235.79 |
| 4 | 1036 | 155,493.1 | 1500.90 |
| 5 | 0 | 0 | 0 |
| 6 | 37 | 145,776.2 | 3939.90 |

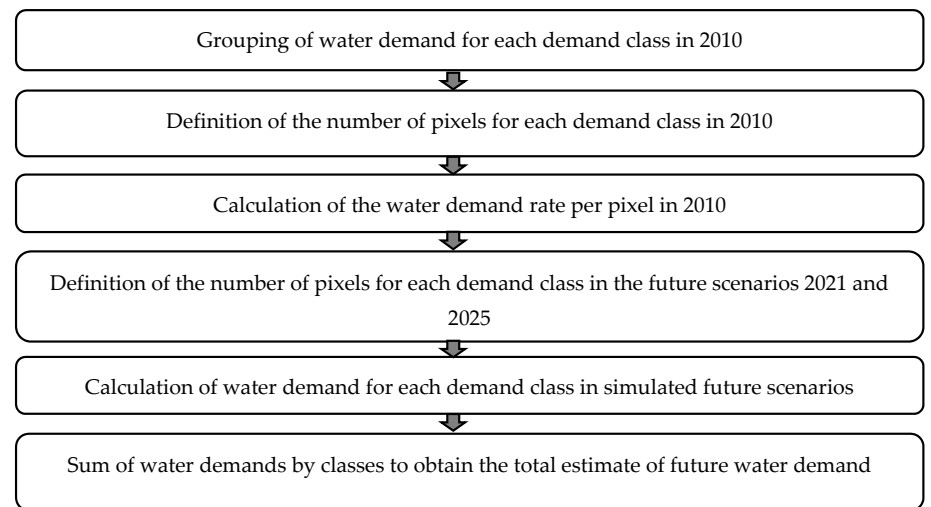

**Figure 6.** Summary of the steps to estimate future water demand.

The demand value of the other census sectors and the value of the distribution losses were added to the demand value to obtain the volume of raw water that will enter the water treatment station (WTS).

The water consumption value of the 91 census sectors with no data was calculated from data from the last demographic census conducted by IBGE, through the multiplication of population data by per capita consumption. The census was carried out in 2010, and it presents the only official data on the population available in Brazil; therefore, the year 2010 was chosen to carry out the calculation.

The results were validated by comparing the results of the water balance carried out by CAGECE for the year 2015.

According to IPLANFOR [31], the population growth of Fortaleza between 2010 and 2040 is estimated at 26%, resulting in an annual growth rate of 0.82%. This rate was included in the simulation of future scenarios of water consumption.

## 5. Results

The water consumption maps of Fortaleza for 2009 and 2013 are illustrated in Figure 7. The most remarkable increases in water demand were detected in the eastern part of the city. According to Dantas [32], the eastern neighborhoods, e.g., Guararapes, Cidade dos Funcionários, and Sabiaguaba, have expanded with the construction of upper middle-class housing, beach clubs, and hotels. This expansion may be one of the reasons for the increase in water consumption in this area.

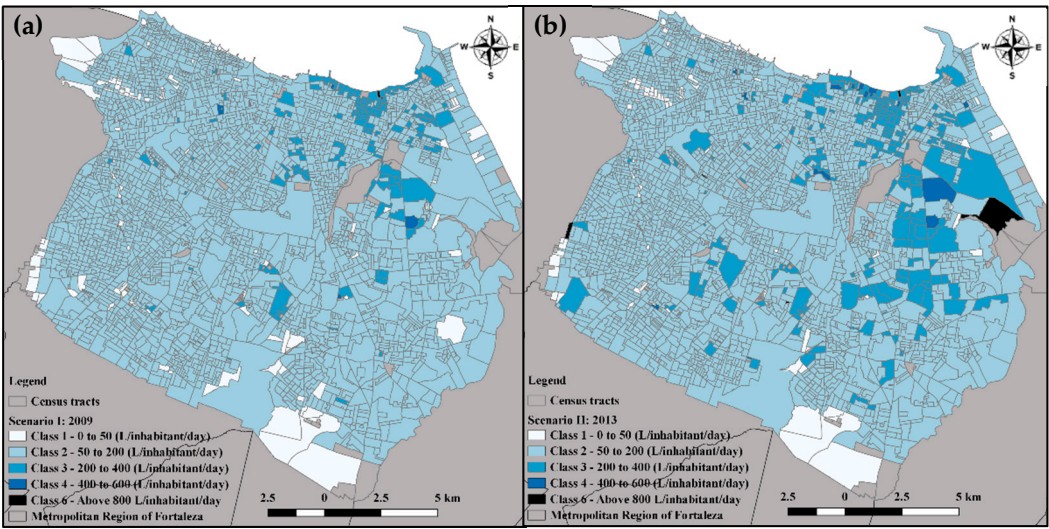

**Figure 7.** (**a**) Average urban water consumption in Fortaleza (L/inhabitant/day) during 2009. (**b**) Average urban water consumption (L/inhabitant/day) during 2013.

Table 3 shows the percent of consumption change between 2009 and 2013. Classes 2 (50 to 200 L/inhabitant/day), 3 (200 to 400 L/inhabitant/day) and 4 (400 to 600 L/inhabitant/day) present an elevated consumption change. Most areas have an average consumption of 300 L/inhabitant/day (class 3), corresponding to an urban water demand of about 735,655,500 L/day.

Figure 8b represents water consumption in 2017 if the consumption patterns from 2009 to 2013 were maintained. The drought period (2013–2017) may have caused a reduction in micro-metering values, Figure 8a. This is demonstrated in Table 1, which shows a water consumption reduction from 2013 to 2017, despite increases in the estimated population.

**Table 3.** Transition matrix.

| Class in 2009 | Class in 2013 | Change Percentage |
|:---:|:---:|:---:|
| 1 | 2 | 24.27% |
| 1 | 3 | 0.10% |
| 2 | 1 | 0.23% |
| 2 | 3 | 9.63% |
| 2 | 4 | 0.04% |
| 2 | 6 | 0.70% |
| 3 | 2 | 7.83% |
| 3 | 4 | 14.01% |
| 4 | 3 | 33.14% |

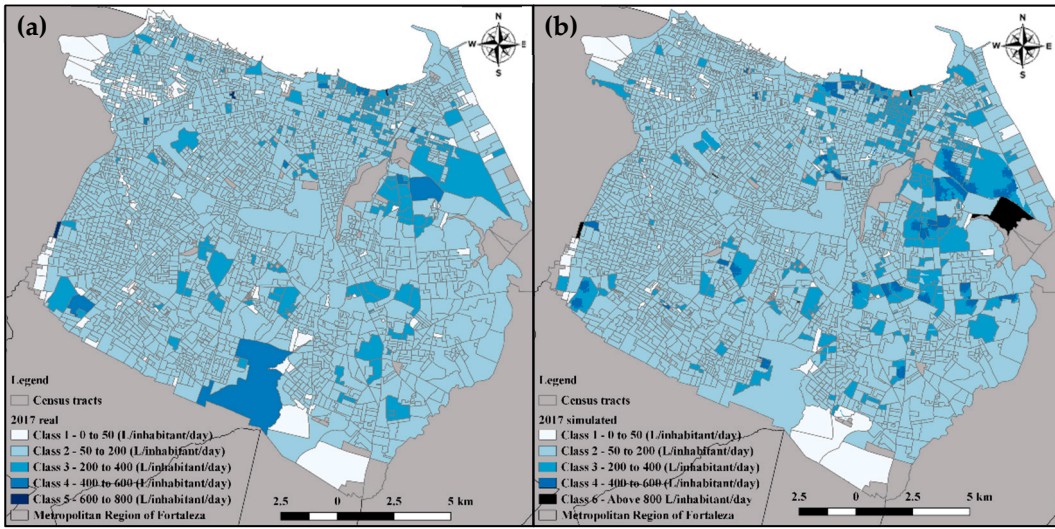

**Figure 8.** (**a**) Average urban water consumption in Fortaleza (L/inhabitant/day) during 2017. (**b**) Simulated average urban water consumption (L/inhabitant/day) for 2017.

The water volume (m³) required to supply Fortaleza in the simulated scenario would be 2.76% higher than the required volume, Table 4. This difference is probably due to the drought conditions and the consequent increase of water consumption tariff that occurred in this period.

**Table 4.** Increase in demanded urban water volume (2017 real × 2017 simulated).

| Water Volume (m³) | | Difference |
|:---:|:---:|:---:|
| 2017 real | 2017 simulated | |
| 285,810,829.3 | 293,700,133.9 | 2.76% |

The parameters used in the urban water demand simulation models are presented in Table 5.

The lowest value of fuzzy similarity between the difference maps was 0.99 for the validation, using 11 × 11 exponential window decay function. This result represents a high similarity between simulated and observed maps, since the minimum requirement for a good level of compatibility is 0.4 [33].

For validation using constant decay function in multiple windows, 3 × 3 and 5 × 5 pixels windows were analyzed. The simulation model obtained a fuzzy similarity equal to 1 (100% similarity). Fuzzy similarity index ranging from 0.45 to 0.50 for windows with sizes from 3 × 3 to 5 × 5 indicates an adequate model performance [29,34].

**Table 5.** Parameters used in the adjustment and execution of the urban water demand simulation model.

| Class in 2009 | Class in 2013 | Average Patch Size (ha) | Variance (ha) | Isometry |
|---|---|---|---|---|
| 1 | 2 | 14.64522059 | 23.46246257 | 1.328104457 |
| 1 | 3 | 2.1875 | 0 | 1.354679803 |
| 2 | 1 | 4.629807692 | 4.357588981 | 1.226899778 |
| 2 | 3 | 35.65357143 | 83.71615399 | 1.363150969 |
| 2 | 4 | 4.78125 | 0.044194174 | 1.301474326 |
| 2 | 6 | 45.515625 | 80.89871778 | 1.386848548 |
| 3 | 2 | 6.279411765 | 4.821274349 | 1.257982941 |
| 3 | 4 | 16.95454545 | 37.1232648 | 1.361264205 |
| 4 | 3 | 4.895833333 | 5.091480834 | 1.327692808 |

In addition to the simulation model validation, observed and simulated maps of 2017 were compared through a simple matrix subtraction (raster format), resulting in a similarity degree of 100%.

In order to validate the estimated water demand for future scenarios, the simulated total water consumption for 2015 was compared to the water balance for that year, demonstrating a difference of 2.38%, Table 6. This overestimation is justified by the estimation made for the census tracts with missing data, which adopted 2010 average consumption. This year probably had increased consumption patterns in comparison to 2015, since the last was an atypical year due to drought conditions.

**Table 6.** Data used to validate future water demand estimates.

| Validation for 2015 (m$^3$) | | Difference |
|---|---|---|
| Simulated | Observed | |
| 288,422,145.4 | 281,706,099.5 | 2.38% |

The prospective scenarios are illustrated in Figure 9. Considering a 0.82% annual population growth, the increases in water demand between 2017 (observed) and 2021 and 2025 are 6.45% and 10.16%, respectively. This represents an annual growth rate of 1.27%.

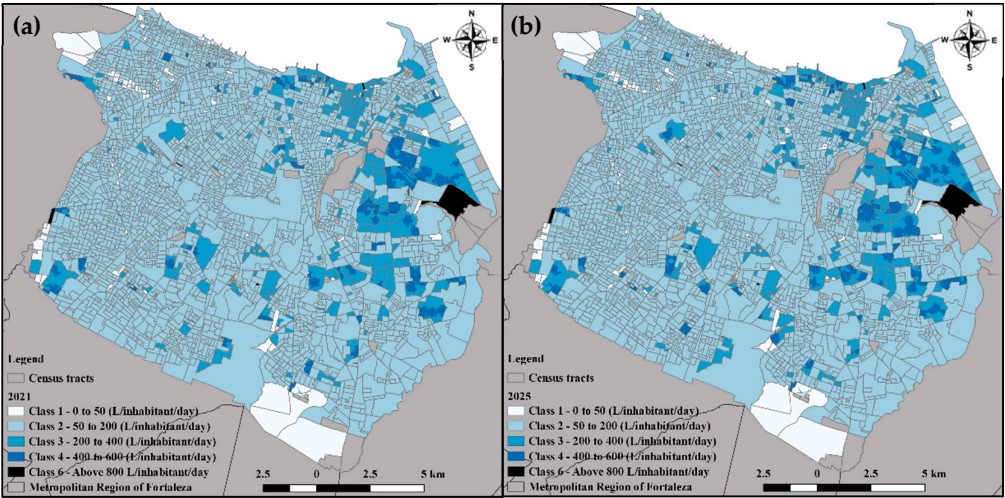

**Figure 9.** (**a**) Simulated average urban water consumption in Fortaleza (L/inhabitant/day) for 2021. (**b**) Simulated average urban water consumption (L/inhabitant/day) for 2025.

Future water demands for 2021 and 2025 were, respectively, 304,246,982.3 m$^3$ and 314,838,965.6 m$^3$. These demands represent the volume that must be supplied by the water company, considering an average loss from distribution networks of 45%. The real and apparent losses of Fortaleza WSS are 20% and 25% respectively.

A linear regression analysis (Figure 10) showed a growing trend of the urban water demand in Fortaleza ($R^2 = 1.0$). In addition, it demonstrates the temporal evolution of urban water demand according to the simulations performed in this study.

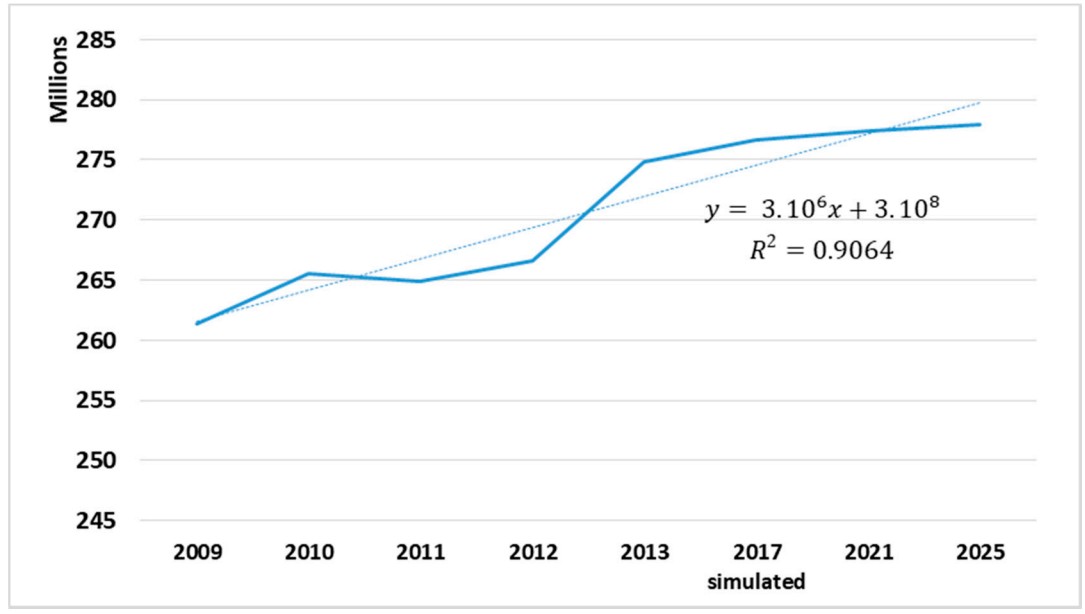

**Figure 10.** Urban demand growth trend in Fortaleza (m$^3$).

## 6. Conclusions

Understanding the increase in water consumption and future trends is fundamental for an adequate administration of urban water supply services. Defining planning strategies using future scenarios can increase water resource system resilience and extreme event preparedness.

In this study, we assessed urban water demand growth in Fortaleza using cellular automata. Water demand thematic maps showed that the city would have had higher consumption rates if the pattern established between 2009 and 2013 was preserved. This was verified with the comparison between the observed and simulated 2017 water consumption maps, where a 2.76% increase was identified. In addition, simulated future water demand (2021 and 2025) was compared to the observed water demand in 2017, indicating a possible increase of 6.45% and 10.16%, respectively.

The transition matrix data obtained through the changes in consumption between 2009 and 2013 indicated that the predominant class has an average consumption of 300 L/inhabitant/day. This information would be useful for strategic planning and for providing support in defining policies that contribute to the development of a more sustainable city.

**Author Contributions:** Conceptualization, methodology and validation, L.M.d.O.; formal analysis and investigation, L.M.d.O. and S.M.O.d.S.; writing—original draft preparation, L.M.d.O., S.M.O.d.S. and T.M.N.C.; software, L.M.d.O.; writing-review and editing, S.M.O.d.S., F.d.A.d.S.F. and R.L.F.; supervision, S.M.O.d.S. and F.d.A.d.S.F. All authors have read and agreed to the published version of the manuscript.

**Funding:** The research was partially supported by grants from the Conselho Nacional de Desenvolvimento Científico e Tecnológico—Brasil (CNPq) (NEXUS Project, N° 441457/2017-7), Fundação Cearense de Apoio ao Desenvolvimento Científico e Tecnológico (FUNCAP), and Coordenação de Aperfeiçoamento de Pessoal de Nível Superior—Brasil (CAPES).

**Conflicts of Interest:** The authors declare no conflict of interest.

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
