# Peer review of "Forecasting Urban Water Demand Using Cellular Automata"

_water, doi:10.3390/w12072038_

Round 1

Reviewer 1 Report

This is an interesting paper, which propose a way to forecast urban water demands. Results are encouraging. This referee is happy to see it published at the end. However, this referee finds that methodology has not yet been described sufficiently and this referee cannot recommend for publication at this point. 

How the cellular automata apply to compile data in the Maps 2009 & 2013? Lines 161-172 give a general description supported with Figure 3 showing the graphical relationship. This referee would like to see how the cellular automation pattern is defined? Please confirm the states in each cell are the consumption classes 1-6. How large of area that one pixel represents? How the Euclidean distance is determined from the mapped cells in the two maps, so that the weights of evidence can be determined. This referee suggest to illustrate them with Equations. 

Reviewer 2 Report

This is an interesting and well written paper which fits the aims of the Journal.

I have only a concern about the "Introduction" where no word is spent about the very important issue of leak reduction for a sustainable water resources management. I'd suggest the Authors to include some comments about this very crucial topic. There are several aspects that could be taken into account. As an example, the use of District Metered Areas (DMA), leak reduction by controlling pressure by means of pressure reducing valves (PRV), and innovative techniques for fault detection could be considered. In my opinion, a 2020 paper where the water demand management is addressed with no attention about the mentioned crucial topics would be less effective.

As an example of papers about the use of innovative techniques for fault detection, paper [a], recently published on Water, could be taken into account (of course, it is not compulsory to cite this paper).

[a] Meniconi, S., Brunone, B., and Frisinghelli, M. (2018). On the role of minor branches, energy dissipation, and small defects in the transient response of transmission mains. Water, 10, 187, 246-262.

Finally, I'd suggest to check the text (some minor errors could be eliminated: e.g., 

row 59: errata "...we used use of ... corrige "we used..."

row 62: errata "article.." corrige "article."

row 344: errata "preparedness ." corrige "preparedness." 

Round 2

Reviewer 1 Report

I am pleased to have the revised manuscript of this work. Thank you for author’s revision. Most of my concerns are addressed reasonably. However, a core concept has not yet been clarified sufficiently. The descriptions (lines 176-185&208-213) to point 5 (How the Euclidean distance is determined from the mapped cells in the two maps, so that the weights of evidence can be determined. This referee suggest to illustrate them with Equations.) are too general. (1) Regarding this point, how the closet distance between class cells is defined? Suppose there are n-th categorical map cells in 2009, 1,2,3…n, the Euclidean distance in the Euclidean plane of map cells (2 dimensions in this paper?) is d(ij)=sqrt{(xj-xi)^2+(yj-yi)^2}, i,j=1,2,3…n. x and y for a cell on map needed to be defined in the text. (2) How the distance map is defined with respect to d(ij)? It needs to be detailed. (3) It seems to this referee that “distances” are determined for all cells (areas?) but not limited to the cells of highest changing potential. Is it the cases? How the highest cell state changing potential is defined? (4) How the probability is determined on the transition probability map? It is noted that a reference is given. It still needs to explain how the method is applied to the case described in this paper. How the probability P(ij) relates to d(ij)?

Besides, for clarity, unit of water consumption (i.e. L/inhabitant/day) should be included in caption of Figures 7-9.

Author Response

This manuscript is a resubmission of an earlier submission. The following is a list of the peer review reports and author responses from that submission.

Round 1

Reviewer 1 Report

The manuscript’s topic of forecasting urban water consumption is a highly important aspect for water management professionals and scientists all over the world due to ongoing urbanisation processes and an increasing pressure on water resources. The authors take the Brazilian city of Fortaleza as a case study for which they apply a forecasting study using a “cellular automata” method to forecast urban water demand for 2021 and 2025.

While the topic of the study is highly relevant, the manuscript in its current form does not provide sufficient information to understand the methodology and hence evaluate the reliability of the results. Please consider my comments below:

Major issues

  • Though I have experience in water demand modelling, especially in applying linear and non-linear multiple regression as well as artificial neural networks, the methods presented in this paper are rather new to me, so I was kind of curious about how the authors conducted their study. However, even after reading the methods section multiple times I do not fully understand the procedure. The descriptions in sections 4.1 and 4.2 are too short – I actually have no idea how the basic mechanism of the model works. I suggest that the authors include some equations or even graphical depictions of their process in order to clarify their procedure. For instance, I do not understand how the static and dynamic variables (table 1) are incorporated into the process, what the “Dinamica EGO model” is and how the “Cellular Automata” process works. The latter aspect is pretty important as this method is mentioned in the manuscript’s title but despite two paragraphs (lines 164-172), this process is not described. Furthermore, the validation procedure using a fuzzy similarity index sounds interesting, but how this specifically works is not presented clearly enough, unfortunately.
  • The data on which the study builds upon seems to be limited. Though this is a typical problem in many settings (data scarcity), only using two time steps (2009 and 2013) for model calibration may be too little. As it seems, there is more “aggregated” data on water demand available, as the authors use the 2015 water demand to validate their results. Maybe the authors can think about incorporating a longer time series of these aggregated data into their analysis to calibrate the model over longer time period – I don’t know if the available date would allow this.
  • In addition, the authors want to use two scenarios to forecast future water demand. But what they actually do is, they just calculate the water demand for two years in the future (2021 and 2025). This is not what scenarios should do: The core idea of scenarios is to create different possible futures based on the current conditions. Hence, the authors may think about defining two scenarios, one worse-case and a best-case scenario which they use to forecast e.g. the 2025 conditions. The way it is currently implemented, I would not say that these are scenarios because the authors only use rough population estimates and multiply these with their water demand which they figured out.

Minor issues

  • Abstract is missing some concluding remarks. What does the research provide for local actors?
  • Lines 36-39: Does per capita water consumption really increase? Do you refer to the South American case here? In Europe for instance, we see a clearly decreasing trend of per-capita consumption since years.
  • Lines 46-49: Here you say that there are already studies that looked into water consumption. So, what did these studies ignore and why is your study now providing new information?
  • Lines 67-69: What you present here is reasonable, but you cite Xu et al. 2018 which is a paper on land use change in China, so I wonder where you find this information in that paper?!
  • Lines 71-76: The arguments you present here are taken over from Suhartono et al. 2018 almost “word by word”. Since Suhartono et al. 2018 present these arguments in their introduction section without giving any references for this, I would suggest that you include more references to support your statements because the way it is now, it is not very reliable.
  • Lines 77-78: Reference for this statement is needed.
  • Lines 99-102: Reference for this statement is needed.
  • Line 105: What is “hm3” kind of unit? Do you mean “km3”?
  • Line 123: Did you collect the data for the entire period between 2009 and 2017 or only for certain years? Your analysis seems to be built only on certain years, right? If more data is available, you could use this to enhance the model calibration process.
  • Lines 144-146: What is the “Dinamica EGO simulation model”? How does it calculate the probability of class changes?
  • Lines 153-154: How do you calculate the annual time step and what is an “ergodic matrix”?
  • Lines 164-172: Here you explain the “cellular Automation rule”, but it is hard to understand what this method does. You may think about presenting the major processes therein as equations and maybe even give a visual impression of how this method works because from your text, I do not have a clear idea.
  • Section 4.2.: This section is intend to present the core method of the study but from my point of view, there is a need to expand this section as the text in its current form does not provide enough information to reproduce the method. Please think about expanding this section or even splitting it into a “model calibration” and a “validation” section.
  • Lines 219-220: Why may have the drought had an impact on “micro-metering values”?
  • Lines 247-252: Why do you compare your model to the 2015 water consumption – I thought your validation year is 2017? If you have more years available for aggregated demand values, you could use this data for model calibration.
  • Line 266: Why do you assume an average loss from the distribution network of 45% when the current losses are between 20 – 25%?
  • Figure 6 & Lines 268-270: Here you present the growing urban water demand which is a reasonable result, as your major input variable is the population which is growing over time. So, in the end the water demand you simulate is primarily driven by population growth, so the trend you present in this figure is not surprising.

Reviewer 2 Report

The paper aims to identify the changes in water consumption in the city of Fortaleza using ‘Dinamica Ego software’, that is a platform for modelling the environmental dynamics using geomatic techniques. It seems a trivial application of the software package without a proper critical analysis.

For the publication of the paper I suggest the following:

2) Improve paragraphs 1 and 2 where the same statements are repeated several times. I suggest to deepen the bibliographic analysis on the use of geomatic techniques and in particular of the 'Dinamica Ego' software, which is the basis of this work

1) Better justify the choice of the two sample years (2009 and 2013) by providing information and data on the trend of demand in the other years of that period. It would be interesting to verify the results by choosing two different sample years. Also, is it possible to select a number of years greater than 2 as a sample?

2) provide a greater theoretical basis for the use of the various components of the software.

Moreover:

Pag.1 row 22 – change ‘data was’ with ‘data were’. ‘Data’ is  plural!

Pag.3 fig.1 – Improve the resolution of this figure

Pag.3 row 111 – change ‘interregional waters’ with ‘interregional storage system’ or something like that. It is meaningless to say 'interregional waters have a total capacity'.

Pag.3 rows 116-117 – change ‘the WSS is composed of capture, adduction, water  treatment, reservation, pumping, and distribution network structures’ in ‘the WSS is composed of works for water captation, adduction, treatment, storage, pumping and distribution’.

Pag.3  row 120 – ‘geotechnology techniques’ ?? Perhaps ’geomatic techniques’

Pag.3 row 123 – ‘micro-metering data’, what do you mean? Isn’t it a usual water metering of user demand?

Pag.4 fig.2 – Improve the resolution of this figure

Pag.4 – row 128 – ‘AC-based’ ?. Please, explain. Does the acronym stand for?

Pag.4 – Remove Table 1 - It is not necessary to insert a table for defining the static and dynamic variable.

Pag.4  row 153-154 Please, explain better what is the difference in considering  ‘full period’ and ‘annual time step’

Pag.4 –row 154. ‘real number Eigenvalues and vectors’ ?? Perhaps: ‘real eigenvalues and eigenvectors’

Pag.5  row 155. Please, explain what 'the weights of evidence' are? Are they connected to the Markov chain probability matrix?

Pag.5 row 157 – ‘line generalizing algorithm’ , please, describe this algorithm and insert a reference

Pag.5  row 159 – change Cramer (V) with ‘Cramer’s V (V)’

Pag.5 row 165 – the sentence is not clear, perhaps a comma is missing.

Pag.5  row 176-180. Please, explain better, this description is confusing. Then, 11x11…..,1x1, 2x3,5x5,7x7,9x9 what? Pixels, I suppose.

Pag.5 – Paragraph 4.3 –  Does water demand forecast start from 2010? Why this choice?

Pag.6 row 194 – ‘missing data (n=91)’ ???

Pag.6 row 195 – IBGE, please explain the acronym for not Brazilian people.

Pag.7 Figure 4 – The results do not appear very good!